# A Novel Low-Rank Embedded Latent Multi-View Subspace Clustering Approach

**DOI:** 10.3390/s25092778

**Published:** 2025-04-28

**Authors:** Sen Wang, Lian Chen, Zhijian Liang, Qingyang Liu

**Affiliations:** School of Science, East China Jiaotong University, Nanchang 330013, China; 2023088085410004@ecjtu.edu.cn (L.C.); 2024088085410001@ecjtu.edu.cn (Z.L.); 2023088085410003@ecjtu.edu.cn (Q.L.)

**Keywords:** low-rank embedding, multi-view learning, heterogeneous feature fusion, latent space learning

## Abstract

Noises and outliers often degrade the final prediction performance in practical data processing. Multi-view learning by integrating complementary information across heterogeneous modalities has become one of the core techniques in the field of machine learning. However, existing methods rely on explicit-view clustering and stringent alignment assumptions, which affect the effectiveness in addressing the challenges such as inconsistencies between views, noise interference, and misalignment across different views. To alleviate these issues, we present a latent multi-view representation learning model based on low-rank embedding by implicitly uncovering the latent consistency structure of data, which allows us to achieve robust and efficient multi-view feature fusion. In particular, we utilize low-rank constraints to construct a unified latent subspace representation and introduce an adaptive noise suppression mechanism that significantly enhances robustness against outliers and noise interference. Moreover, the Augmented Lagrangian Multiplier Alternating Direction Minimization (ALM-ADM) framework enables efficient optimization of the proposed method. Experimental results on multiple benchmark datasets demonstrate that the proposed approach outperforms existing state-of-the-art methods in both clustering performance and robustness.

## 1. Introduction

With the continuously emerging large amounts of data and information from various perspectives, data from different viewpoints often exhibit high degrees of complementarity [1]. Due to the consistency between different perspectives, learning from a single view is insufficient to capture full information from the data [2,3,4]. This is particularly critical in cross-view learning, which allows one to effectively integrate heterogeneous data such as images, text, and audio towards promoting mutual understanding and collaborative learning across different views [5]. To alleviate this, multi-view learning by integrating data from various perspectives is presented to uncover latent patterns to enhance the performance and generalization ability of algorithms. By treating the subspace representations of different views as tensors and integrating with a low-rank constraint, we can effectively capture higher-order correlations in multi-view data, to reduce redundancy and improve the clustering accuracy [6,7]. In multi-view learning, low-rank embedding methods facilitate the processing of high-dimensional data, reduce the impact of noise, and improve the robustness and generalization capabilities of the model. Meanwhile, shared representations and kernel methods are integrated to further enhance the performance of cross-modal clustering and effectively address nonlinearity and noise issues in the data [8,9].

In recent years, multi-view learning has encompassed a variety of methods for integrating information from different perspectives. In the early stages, traditional approaches, such as co-training, have significantly improved model learning performance by leveraging complementarity between multiple views [10,11]. Locality-preserving projection (LPP) is a nonlinear dimensionality reduction technique that aims to preserve the local structure of the data by minimizing the reconstruction error between neighboring data points and effectively captures nonlinear relationships in the data, thus improving the robustness of the model to noise and incomplete labels [12]. Low-rank representation (LRR), on the other hand, enables one to enhance the accuracy of data representation in low-dimensional space by capturing the global structure of the data. Recently, improved variants of LRR, such as the graph-based LRR and weighted LRR, have been proposed to enhance the robustness when dealing with high-dimensional noisy or incomplete data [13]. Latent space is a core concept in representation learning, which can be used to map high-dimensional data to low-dimensional latent representations and to capture the deepest features in the data [14,15]. Moreover, the optimization techniques such as Variational Autoencoders (VAEs) and Conditional Generative Adversarial Networks (cGANs) have been proposed to enhance the expressiveness of the latent space.

Although the LRR model has demonstrated the advantages in handling sparse noise, this model is also limited in practical applications due to the sensitivity to non-sparse and structured noise that significantly restricts robustness in real-world scenarios [16]. Moreover, the traditional LRR methods primarily focus on capturing the global structure of data while overlooking the preservation of local geometric features [17]. Despite these advancements, existing methods still suffer from the following key challenges: (1) Each view in multi-view data may contain noise and outliers, which can negatively impact the learning process, thereby reducing the accuracy and robustness of the model; (2) Different views often exhibit distinct data distributions, which are crucial to identify and leverage the shared latent structure between multiple views; (3) Multi-view learning typically involves processing high-dimensional data, and the computational complexity of the model tends to increase with the number of views and data dimensions. Addressing these limitations is necessary for the development of more robust frameworks with strong theoretical foundations to simultaneously tackle noise adaptability, local-global structure preservation, and computational efficiency.

Multi-view learning often faces challenges from inherent data redundancy and noise contamination, which propagate through affinity matrices and degrade representation learning performance. To alleviate this issue, we assume that multiple heterogeneous views originate from a shared latent representation. Based on this assumption, we propose a novel Low-Rank Embedded Latent Multi-View Subspace Clustering (LRE-LAMVSC) framework. First, due to the capability to retain global discriminative structure in reduced space by the low-rank embedding method [18], we integrate this method into latent space learning by decomposing the global subspace through a projection matrix, effectively capturing cross-view consistency while suppressing noise propagation. Then, we apply the latent representation learning mechanism to unify different feature modalities (e.g., visual and textual data) into a common subspace and to mitigate feature mismatches caused by inter-view differences while preserving complementary information. Finally, our comprehensive experimental results demonstrate that the LRE-LAMVSC achieves superior clustering accuracy and robustness by comparing to the state-of-the-art methods while maintaining computational efficiency for practical deployment.

The main contributions of this paper are as follows:(1)By leveraging low-rank embedding to model the shared subspace structure of multi-view data, our approach effectively suppresses noise and redundant information that allows us to enhance the robustness in handling outliers and noisy environments;(2)We can map information from different views into a unified semantic space through a latent multi-view learning fusion mechanism. This process preserves the unique characteristics of each view while strengthening inter-view consistency;(3)We have designed a cross-view projection mapping mechanism that transforms the original high-dimensional heterogeneous data into a low-rank latent space. This allows the separation of shared subspace and view-specific information, enabling a more precise capture of the consistent structure across multiple views.

The structure of this thesis is as follows. Section 2 provides a brief review of the key algorithms related to low-rank embedding and latent spaces. Section 3 introduces the proposed LRE-LAMVSC algorithm along with the optimization strategy. In Section 4, we conduct experiments on real-world datasets to demonstrate the superiority of the proposed algorithm. Finally, Section 5 concludes the paper with a summary of the research findings.

## 2. Related Works

Recent advancements in multi-view learning have led to the development of various approaches that have been effectively applied across different practical scenarios. In the field of multi-task learning, Cai et al. proposed the Guided Multi-Task Learning (GMT) framework by balancing task detection and Re-ID sub-tasks conflicts using the GMHL module and BMOF method, which enabled them to enhance the efficiency and accuracy of end-to-end person search [19]. Fu et al. have proposed a robust modeling framework that integrates manifold regularization by learning a low-rank coefficient matrix to capture the global structure of the original data, aiming to reduce the negative impact of outliers and to enhance the ability to interpret the global structure of industrial data containing anomalies [20,21]. Nikita et al. have proposed a novel gradient-based optimization method (MAMGD) using exponential decay by incorporating mechanisms that include exponential decay, adaptive learning rate, and discrete second-order derivatives, which improved convergence speed and stability across multiple function optimization tasks and neural network training scenarios [22]. Zheng et al. have introduced a multi-view subspace clustering method with feature concatenation and graph regularization. This method improved the model performance by exploiting consistency information across multiple views, while enhancing the robustness of the algorithm by leveraging both the consistency and complementarity of the data from different views [23]. Maria et al. have proposed a multi-view low-rank sparse subspace clustering method that learns a joint subspace to represent the information from all views. This method jointly learned an affinity matrix constrained by sparsity and low-rank assumptions to capture the shared information across views. Zhang et al. have addressed the computational cost associated with low-rank structure learning in multiple views by proposing a fast multi-view subspace clustering method. This approach decomposes the data into two smaller factor matrices, which reduces the computational cost of solving the SVD problem while preserving the underlying low-rank structure of the views [24]. Low-rank linear embedding for robust clustering employs low-rank representation, showing excellent performance in single-view clustering. Similar to SCBSV, it applies a low-rank representation to each view to achieve optimal clustering results [25]. Wang et al. proposed a popular method based on dual consistency to guide incomplete multi-view clustering to achieve consensus representation through reverse regularization by recovering incomplete view data and, thus, to balance the importance between tasks and improve the clustering accuracy [26]. Multi-view learning methods have made significant advancements and have been widely applied across various practical domains. Numerous approaches have significantly enhanced the robustness and computational efficiency of multiview data modeling by integrating techniques such as low-rank representation, graph regularization, and sparsity constraints.

Traditional multi-view learning methods enhanced modeling performance by integrating same-type information from different perspectives within the same dataset (e.g., images and text), but they were constrained by explicit feature fusion and homogeneous data representations [27]. Unlike traditional multi-modal learning, the binary multi-view clustering method presented in [28] is capable of simultaneously processing heterogeneous data from different modalities (e.g., text, speech, etc.) and, by modeling the complex nonlinear relationships between these modalities, it achieves deeper semantic understanding and improved generalization capabilities [29]. Recently, research on the latent multi-view subspace clustering has mainly focused on integrating features from different perspectives to uncover low-dimensional representations of the data, with the aim of improving clustering accuracy and addressing issues of heterogeneity and data inconsistency across views. Zhang et al. proposed a new multi-view subspace clustering model based on latent representation [29]. Compared to the existing methods, this model simultaneously searches for latent representations while exploiting the complementary information across multiple views. This enables the model to represent data in a more comprehensive manner than a single view, making the subspace representation more precise and robust. Xue et al. introduced an unsupervised multi-view dimensionality reduction method based on a dual-layer latent space learning approach. By learning in layers, this method addresses the differences between views and utilizes both shared and private information to preserve the features of each view. The second layer employs joint matrix factorization to fuse the different views into a low-dimensional representation, enhancing both flexibility and robustness [30]. Furthermore, Yao et al. proposed a novel tensor-based incomplete multi-view clustering method that improved missing-view completion with dual tensor constraints and addressed the feature degeneration problem with an adaptive fusion graph learning strategy [31]. Huang et al. have proposed a new multi-view joint learning framework that integrates multiple similarity matrices and applies nuclear norm regularization to learn reliable similarity matrices and low-dimensional latent representations, aiming to improve the robustness over noises and outliers [32]. Many researchers have proposed the robust multi-view spectral clustering method, which constructs a shared transition probability matrix through low-rank and sparse decomposition to reduce the impact of noise in multi-view data and improve multi-view clustering performance [33]. Unlike the RMSC method, the diversity-induced multi-view subspace clustering method reduced data redundancy by integrating subspace representations from different views into a single affinity matrix [34]. The Multi-view Intact Space Learning (MISL) algorithm focuses on integrating multi-view information and learning the latent complete representation, while the multi-instance clustering algorithm focuses on the multi-view analysis and anomaly detection of sensor data in smart building system [1,27]. Therefore, research on the latent multi-view subspace clustering primarily focuses on integrating features from different perspectives to uncover low-dimensional representations of the data, aiming to improve the clustering accuracy and to address the issues of heterogeneity and data inconsistency across views. To this end, various methods have been proposed by leveraging the multi-view information, to reduce the noise interference, to enhance the robustness of models, and, finally, to effectively improve clustering performance and model stability.

The latent rank embedding multi-view learning has been effectively integrated with information from different views by using low-rank representations, which helps to reduce the negative impact of noise and outliers, towards improving the robustness of dimensionality reduction and clustering on multi-view data. Zhao et al. proposed the cross-lingual font style transfer with full-domain convolutional attention (FCAGAN) model, which enabled cross-lingual font style transfer with a small number of samples. Unlike traditional methods, FCAGAN can transfer the font style of one language to another [35]. Zhou et al. proposed an end-to-end robust clustering method (RCLR) that combines low-rank linear embedding with k-means clustering, capable of simultaneously learning sparse coefficients and spatial projection matrices while capturing both global and local structures [36]. This method integrates clustering, dimensionality reduction, low-rank representation, and local property preservation into a unified model, effectively avoiding the error accumulation issue encountered in traditional two-stage frameworks. Li et al. have introduced a low-rank embedding-based method for single-view and multi-view learning, which integrates multiple views from drug and protein data to improve the accuracy of interaction prediction [37]. Meng et al. have proposed a multi-view low-rank preserving embedding method, which effectively integrates multi-view information by minimizing the inconsistency between each view and a central view while preserving the low-rank reconstruction relationships. Unlike existing methods, this approach automatically assigns appropriate weights to each view, eliminating the need for manual parameter tuning [8]. The low-rank tensor constrained multi-view subspace clustering method employs the low-rank tensor decomposition techniques to enforce consistency in high-order data [38]. Additionally, it integrates complementary information from multiple views to enhance the efficiency of data processing. However, the multi-modal sparse and low-rank subspace clustering method constructs a unified model of data from different views to obtain a shared representation [39], thereby enhancing the efficiency of multi-view learning. Latent rank embedding multi-view learning integrates information from different views, reducing the impact of noise and outliers, thereby improving the robustness of dimensionality reduction and clustering. Related methods include low-rank embedding-based linear dimensionality reduction and multi-view low-rank preserving embedding, which improve data integration by automatically optimizing the view weights and reducing inconsistency between views, while eliminating the necessity for manual parameter tuning.

## 3. Proposed Approach

In this section, we first present a novel representation method, i.e., the Low-Rank Embedding (LRE), which preserves the low-rank reconstruction relationships among samples in various ways. Then, by integrating with the latent space learning, we further propose a Low-Rank Embedding-based Latent Multi-View Representation Learning (LRE-LAMVSC) method. This approach models the shared subspace of multi-view data using the structure of low-rank embedding and then utilizes a fusion mechanism in latent multi-view learning to map information from different views into a unified semantic space. Finally, we design a cross-view projection mapping that maps high-dimensional data into the low-rank latent space to effectively capture the consistency structure across multi-view data. Figure 1 illustrates the structure of the LRE-LAMVSC algorithm and Table 1 shows the explanations of the key notations throughout this work.

### 3.1. Low-Rank Embedding

The main objective of the low-rank embedding is to improve the robustness in addressing the linear dimensionality reduction problem related to characteristic damage and outliers, towards enhancing the robustness. To achieve this, low-rank embedding uses the $l_{2,1}$ norm as the fundamental measure for error reconstruction. The objective function of the LRE can be expressed in the following formula [40]:(1)$$\begin{array}{c} \min_{Z,P}\mathrm{rank}(Z)+\lambda {∥P^{T}X-P^{T}\mathrm{XZ}∥}_{2,1}\\ s.t.P^{T}P=I. \end{array}$$

In Equation (Equation 1), $X=[x_{1},x_{2},\ldots ,x_{n}]$ contains the input raw data, Z is the low-rank matrix, and P is the orthogonal subspace. λ is the regularization parameter.

The objective of Equation (Equation 1) is to find the optimal low-rank reconstruction and projection, with the primary goal to model the specific damage and outlier issues of the samples. Therefore, the main problem can be addressed by replacing the relevant lower bounds with a rank function, for which we introduce the kernel norm minimization as follows:(2)$$\begin{array}{c} \min_{Z,P}{∥Z∥}_{*}+\lambda {∥P^{T}X-P^{T}\mathrm{XZ}∥}_{2,1}\\ s.t.P^{T}P=I. \end{array}$$

In Equation (Equation 2),  ${∥\cdot ∥}_{*}$ denotes the nuclear norm of a matrix. The ${∥\cdot ∥}_{2,1}$ norm exhibits superior robustness compared to the Frobenius norm, making the nuclear norm a more stable choice when dealing with outliers or corrupted sample data.

### 3.2. The Latent Multi-View Representation Based on Low-Rank Embedding

In real-world data, each view often carries specific physical significance but suffers from the susceptible to noise. To address this issue, we propose a novel low-rank error reconstruction relationship to improve the embedding of the given views. This approach not only preserves the low-rank error reconstruction relationships between samples in the original space, but also ensures their validity in a latent space. By minimizing the spatial reconstruction function, we maintain the low-rank relationships between samples in a stable manner:(3)$$\min_{Z^{\left(v\right)},P^{\left(v\right)}}\sum_{v=1}^{V}{∥P^{\left(v\right)}X^{\left(v\right)}-Z^{\left(v\right)}∥}_{F}^{2}+\lambda \sum_{v=1}^{V}{∥P^{\left(v\right)}∥}_{*}.$$

In Equation (Equation 3),  $X^{\left(v\right)}=\left[X_{1}^{\left(v\right)},\ldots ,X_{n_{v}}^{\left(v\right)}\right]\in R^{d_{v}\times n}$ denotes the data in the *v*-th view. Let $d_{v}$ represent the dimensionality of the -th view and $P^{\left(v\right)}$ the projection matrix for the -th view, which maps $X^{\left(v\right)}$ to $R^{\left(v\right)}$. The matrix $Z^{\left(v\right)}$ serves as the shared low-dimensional embedding across different views. This model effectively reduces the dimensionality and denoises the data, while maintaining strong structural consistency and significantly enhancing robustness and cross-view correlation. In Equation (Equation 1), there exists a trivial solution for $P^{\left(v\right)}=0$ and $Z^{\left(v\right)}=0$ that leads to a lower bound of zero. To resolve this, an additional constraint $P^{\left(v\right)}{P^{\left(v\right)}}^{T}=I$ is introduced, along with a regularization term involving the Frobenius norm. The resulting formulation is as follows:(4)$$\begin{array}{c} \min_{Z^{\left(v\right)},P^{\left(v\right)}}\sum_{v=1}^{v}({∥P^{\left(v\right)}X^{\left(v\right)}-Z^{\left(v\right)}∥}_{F}^{2}+\lambda {∥P^{\left(v\right)}∥}_{F}^{2})\\ s.t.P^{\left(v\right)}{P^{\left(v\right)}}^{T}=I. \end{array}$$

Low-rank embedding is highly sensitive to noise and missing values, which often results in outputs deviating from the true underlying structure. Furthermore, the limited interpretability and poor adaptability to real-world tasks hinder the ability to produce embeddings that meet the specific demands of these tasks. To overcome these limitations, combining low-rank embedding with latent space learning can significantly enhance the model generalization ability. This enables the model to capture complex patterns in the data while preserving both local and global structural information.

Low-rank embedding can effectively reduce dimensionality, denoise, and uncover the underlying structure of the data, while latent multi-view learning integrates information from multiple views to enhance the model’s expressiveness, robustness, and generalization capability. By combining low-rank embedding with latent multi-view representation learning, the strengths of both approaches can be leveraged, resulting in a more efficient low-rank embedding latent multi-view representation learning model. The final objective function can be expressed as follows:(5)$$\begin{array}{c} \min_{Z^{\left(v\right)},P^{\left(v\right)},E^{\left(v\right)}}\sum_{v=1}^{V}\left({∥P^{\left(v\right)}X^{\left(v\right)}-Z^{\left(v\right)}∥}_{F}^{2}+\varphi {∥E^{\left(v\right)}∥}_{2,1}+\lambda {∥P^{\left(v\right)}∥}_{F}^{2}+\gamma {∥Z^{\left(v\right)}∥}_{*}\right)\\ s.t.P^{\left(v\right)}P^{{\left(v\right)}^{T}}=I. \end{array}$$

In Equation (Equation 5),  $X^{\left(v\right)}$ is the input matrix of the raw data, $P^{\left(v\right)}$ is the projection matrix, $Z^{\left(v\right)}$ is the representation learned by the model, and $E^{\left(v\right)}$ is the error matrix obtained after error reconstruction; λ and γ are regularization parameters. This model can improve both the robustness and adaptability of the embedding process, while also demonstrating greater tolerance to noise and missing values. Furthermore, the relevant low-dimensional representations generated through latent space learning can further enhance the robustness and generalization ability of the embedding.

### 3.3. ALM-ADM Optimization

Based on the final objective function derived in Section 3.2, our goal is to identify the low-rank projection space and effective latent representations across different views. Since not all variables in the objective function are jointly convex, we decompose the objective into sub-problems that can be efficiently solved. To tackle these sub-problems, we employ the Augmented Lagrangian Multiplier Alternating Direction Minimization (ALM-ADM) framework method, which incorporates the constraints into the objective function. By introducing the augmented Lagrange multipliers and penalty terms, the method progressively brings the solution closer to satisfying the constraints during optimization. To ensure that the objective function is separable during optimization, we introduce an auxiliary variable, $S^{\left(v\right)}$, to transform the problem into the following objective function:(6)$$\begin{array}{c} \min_{Z^{\left(v\right)},P^{\left(v\right)},E^{\left(v\right)}}\sum_{v=1}^{v}({∥P^{\left(v\right)}X^{\left(v\right)}-Z^{\left(v\right)}∥}_{F}^{2}+\varphi {∥E^{\left(v\right)}∥}_{2,1}+\\ \lambda {∥P^{\left(v\right)}∥}_{F}^{2}+\gamma {∥Z^{\left(v\right)}∥}_{*})\\ s.t.P^{\left(v\right)}{P^{\left(v\right)}}^{T}=I,E^{\left(v\right)}=P^{\left(v\right)}X^{\left(v\right)}-Z^{\left(v\right)},Z^{\left(v\right)}=S^{\left(v\right)}. \end{array}$$

Thus, the augmented Lagrangian function for the above problem takes the following form:(7)$$\begin{array}{c} L=\min_{Z^{\left(v\right)},P^{\left(v\right)},E^{\left(v\right)},S^{\left(v\right)}}\sum_{v=1}^{v}({∥P^{\left(v\right)}X^{\left(v\right)}-Z^{\left(v\right)}∥}_{F}^{2}+\varphi {∥E^{\left(v\right)}∥}_{2,1}+\lambda {∥P^{\left(v\right)}∥}_{F}^{2}+\gamma {∥S^{\left(v\right)}∥}_{*})\\ +\langle Y_{1},Z^{\left(v\right)}-S^{\left(v\right)}\rangle +\langle Y_{2},P^{\left(v\right)}{P^{\left(v\right)}}^{T}-I\rangle +\langle Y_{3},P^{\left(v\right)}X^{\left(v\right)}-Z^{\left(v\right)}-E^{\left(v\right)}\rangle \\ +\frac{\mu }{2}\sum_{V=1}^{V}\left({∥P^{\left(v\right)}{P^{\left(v\right)}}^{T}-I∥}_{F}^{2}+{∥Z^{\left(v\right)}-S^{\left(v\right)}∥}_{F}^{2}+{∥P^{\left(v\right)}X^{\left(v\right)}-Z^{\left(v\right)}-E^{\left(v\right)}∥}_{F}^{2}\right) \end{array}.$$

In Equation (Equation 7),  μ is the penalty parameter and $Y_{1},Y_{2},Y_{3}$ are the Lagrange multipliers. Therefore, the above objective function can be redefined as(8)$$L=\min_{Z^{\left(v\right)},P^{\left(v\right)},E^{\left(v\right)},S^{\left(v\right)}}\sum_{v=1}^{v}\left({∥P^{\left(v\right)}X^{\left(v\right)}-Z^{\left(v\right)}∥}_{F}^{2}+\varphi {∥E^{\left(v\right)}∥}_{2,1}+\lambda {∥P^{\left(v\right)}∥}_{F}^{2}\;+\gamma {∥S^{\left(v\right)}∥}_{*}+\frac{\mu }{2}({∥P^{\left(v\right)}{P^{\left(v\right)}}^{T}-I+\frac{Y_{1}}{\mu }∥}_{F}^{2}+{∥Z^{\left(v\right)}-S^{\left(v\right)}+\frac{Y_{2}}{\mu }∥}_{F}^{2}\;+{∥P^{\left(v\right)}X^{\left(v\right)}-Z^{\left(v\right)}-E^{\left(v\right)}+\frac{Y_{3}}{\mu }∥}_{F}^{2})\right).$$

Therefore, the above problem is unconstrained and can be optimized by fixing the other variables separately, using the following steps:

Step 1, Update $S^{\left(v\right)}$, fix $E^{\left(v\right)}$, $Z^{\left(v\right)}$, $P^{\left(v\right)}$:(9)$$\min_{S^{\left(v\right)}}\gamma {∥S^{\left(v\right)}∥}_{*}+\frac{\mu }{2}{∥Z^{\left(v\right)}-S^{\left(v\right)}+\frac{Y_{2}}{\mu }∥}_{F}^{2}.$$

In Equation (Equation 9),  $S^{\left(v\right)}$ can be computed using Singular Value Thresholding (SVT).(10)$$S_{k+1}^{\left(v\right)}=\Theta_{\frac{1}{\mu }}\left(Z^{\left(v\right)}+\frac{Y_{2}}{\mu }\right).$$

In Equation (Equation 10),  $\Theta_{\left(1/\mu \right)}$ represents the scaling factor in Singular Value Thresholding (SVT).

Step 2, Update $Z^{\left(v\right)}$, fix $E^{\left(v\right)}$, $S^{\left(v\right)}$, and $P^{\left(v\right)}$ as follows:(11)$$L=\min_{Z^{\left(v\right)}}\sum_{v=1}^{v}\left({∥P^{\left(v\right)}X^{\left(v\right)}-Z^{\left(v\right)}∥}_{F}^{2}+\frac{\mu }{2}({∥Z^{\left(v\right)}-S^{\left(v\right)}+\frac{Y_{2}}{\mu }∥}_{F}^{2}\;+{∥P^{\left(v\right)}X^{\left(v\right)}-Z^{\left(v\right)}-E^{\left(v\right)}+\frac{Y_{3}}{\mu }∥}_{F}^{2})\right).$$

Equation (Equation 11) can be further rewritten in trace form and then the derivative can be computed as follows:(12)$$-4P^{\left(v\right)}X^{\left(v\right)}+6Z^{\left(v\right)}-2S^{\left(v\right)}+\frac{2}{\mu }\left(Y_{3}-Y_{2}\right).$$

By setting the derivative to zero, the final solution of $Z^{\left(v\right)}$ is(13)$$Z^{\left(v\right)}=\frac{1}{3}S^{\left(v\right)}+2P^{\left(v\right)}X^{\left(v\right)}-\frac{2}{\mu }\left(Y_{3}-Y_{2}\right).$$

Step 3, Update $P^{\left(v\right)}$ and fix $E^{\left(v\right)}$, $S^{\left(v\right)}$, $Z^{\left(v\right)}$ as follows:(14)$$L=\min_{P^{\left(v\right)}}\sum_{v=1}^{v}\left({∥P^{\left(v\right)}X^{\left(v\right)}-Z^{\left(v\right)}∥}_{F}^{2}+\lambda {∥P^{\left(v\right)}∥}_{F}^{2}+\frac{\mu }{2}({∥P^{\left(v\right)}{P^{\left(v\right)}}^{T}-I+\frac{Y_{1}}{\mu }∥}_{F}^{2}\;+{∥P^{\left(v\right)}X^{\left(v\right)}-Z^{\left(v\right)}-E^{\left(v\right)}+\frac{Y_{3}}{\mu }∥}_{F}^{2})\right).$$

This function can be expanded into trace form and the gradient with respect to $P^{\left(v\right)}$ can be further computed as follows:(15)$$\begin{array}{c} \nabla L\left(P^{\left(v\right)}\right)=2\left(P^{\left(v\right)}X^{\left(v\right)}-Z^{\left(v\right)}\right)X^{{\left(v\right)}^{T}}+2\lambda P^{\left(v\right)}+\mu \left(P^{\left(v\right)}P^{{\left(v\right)}^{T}}-I+\frac{Y_{1}}{\mu }\right)P^{\left(v\right)}\\ +\mu \left(P^{\left(v\right)}X^{\left(v\right)}-Z^{\left(v\right)}-E^{\left(v\right)}\right)X^{{\left(v\right)}^{T}} \end{array}.$$

During the gradient descent process, the objective function can be minimized by updating $P^{\left(v\right)}$ as follows:(16)$$P_{k+1}^{\left(v\right)}=P_{k}^{\left(v\right)}-\eta \nabla L\left(P^{\left(v\right)}\right).$$

In Equation (Equation 16), η represents the learning rate in the gradient descent process and *k* is the iteration count.

Then, the iteration stops when the change in the objective function is smaller than a threshold ε as follows:(17)$$\left|L\left(P_{k+1}^{\left(v\right)}\right)-L\left(P_{k}^{\left(v\right)}\right)\right|<\varepsilon .$$

Step 4, Update $E^{\left(v\right)}$ and fix $P^{\left(v\right)}$, $S^{\left(v\right)}$, $Z^{\left(v\right)}$ as follows:(18)$$L=\min_{E^{\left(v\right)}}\varphi {∥E^{\left(v\right)}∥}_{2,1}+\frac{\mu }{2}\left({∥P^{\left(v\right)}X^{\left(v\right)}-Z^{\left(v\right)}-E^{\left(v\right)}+\frac{Y_{3}}{\mu }∥}_{F}^{2}\right).$$

Let $P^{\left(v\right)}X^{\left(v\right)}-Z^{\left(v\right)}+\frac{Y_{3}}{\mu }=M^{\left(v\right)}$; this problem can be reformulated as follows:(19)$$\min_{E^{\left(v\right)}}\varphi {∥E^{\left(v\right)}∥}_{2,1}+\frac{\mu }{2}\left({∥M^{\left(v\right)}-E^{\left(v\right)}∥}_{F}^{2}\right).$$

The optimal solution to the above objective function can be further expressed as(20)$${\left[E^{*}\right]}_{:,i}=\left\{\begin{array}{cc} \frac{{∥{\left[M\right]}_{:,i}∥}_{2}-\varphi }{{∥{\left[M\right]}_{:,i}∥}_{2}} & \mathrm{if}\;{∥{\left[M\right]}_{:,i}∥}_{2}>\varphi \\ 0 & \mathrm{otherwise} \end{array}\right.{\left[M\right]}_{:,i}.$$

Then, the optimization can be performed through matrix multiplication as follows:(21)$$E^{*}=M\Lambda ,$$

In Equation (Equation 21), $\Lambda =\mathrm{diag}\left[clamp\left(\frac{{∥M∥}_{2}^{\dim=1}-\frac{1}{\mu }}{{∥M∥}_{2}^{\dim=1}},0\right)\right]$.

Update Lagrange multipliers $Y_{1},Y_{2},Y_{3}$ and the penalty parameter μ: (22)$$\begin{array}{cc} Y_{1,k+1} & =Y_{1}+\mu \left(Z^{\left(v\right)}-S^{\left(v\right)}\right), \end{array}$$(23)$$\begin{array}{cc} Y_{2,k+1} & =Y_{2}+\mu \left(P^{\left(v\right)}{P^{\left(v\right)}}^{T}-I\right), \end{array}$$(24)$$\begin{array}{cc} Y_{3,k+1} & =Y_{3}+\mu \left(P^{\left(v\right)}X^{\left(v\right)}-Z^{\left(v\right)}-E^{\left(v\right)}\right), \end{array}$$(25)$$\begin{array}{cc} \mu_{k+1} & =\min(\rho \mu_{k},\mu_{\max}). \end{array}$$

In Equation (25),  ρ is a user-defined balancing parameter. Algorithm 1 outlines the complete optimization process of LRE-LAMVSC. Then, in practical applications, the representation matrix $Z^{\left(v\right)}$ can be randomly initialized to prevent convergence to an all-zero solution. Furthermore, the complexity of LRE-LAMVSC is determined by the dimension of the views. In specific, the computational complexity of the proposed method is $O(k^{2}d+d^{3})$ and $O(n^{3})$ in the matrix update process, where *k* represents the latent dimension and *d* represents the original data dimension. Since the latent representation’s dimension is significantly smaller than that of the original data, the computational complexity can be noted as $O(n^{3}+d^{3})$.
**Algorithm 1:** Optimization for the LRE-LAMVSC.    **Input:**Multi-view matrices $\left\{x^{\left(1\right)},\ldots ,x^{\left(v\right)}\right\}$, hyperparameters λ, γ, φ, and the dimension *k* of latent representation *Z*.    Set $P^{\left(v\right)}=Z^{\left(v\right)}=E^{\left(v\right)}=S^{\left(v\right)}=0$**Repeat:**    update $P^{\left(v\right)}$ according to Problem (9)    update $Z^{\left(v\right)}$ according to Problem (11)    update $E^{\left(v\right)}$ according to Problem (14)    update $S^{\left(v\right)}$ according to Problem (18)    update $Y_{1},Y_{2},Y_{3}$, and μ according to Problems (22–25)**Until:**    $P^{\left(v\right)}P^{{\left(v\right)}^{T}}-I<\varepsilon$    $E^{\left(v\right)}-P^{\left(v\right)}X^{\left(v\right)}-Z^{\left(v\right)}<\varepsilon$    $Z^{\left(v\right)}-S^{\left(v\right)}<\varepsilon$**Output:**    $P^{\left(v\right)},Z^{\left(v\right)},E^{\left(v\right)},S^{\left(v\right)}$

## 4. Results and Discussion

In this section, we evaluate the performance of LRE-LAMVSC by comparing it with several classic multi-view learning methods on four benchmark datasets. First, Section 4.1 provides parameter settings and evaluation indicators. Next, Section 4.2 provides a detailed description of the datasets and comparison methods. Then, comparative studies are conducted in Section 4.3, and comparisons with five state-of-the-art methods are summarized in Section 4.4.

### 4.1. Parameter Settings and Evaluation Indicators

In the experiment, we used four parameters to evaluate the performance of our model. The parameters λ,γ,φ,μ were set as follows for the four datasets MSRCV1, Reuters, ORL, and BBCSport: $\left\{\lambda =1.175,\gamma =1.5,\varphi =1\times 10^{-4},\mu =1\times 10^{-6}\right\}$, $\left\{\lambda =0.8,\gamma =1.5,\varphi =1\times 10^{-6},\mu =1\times 10^{-4}\right\}$, $\left\{\lambda =0.8,\gamma =1.5,\varphi =1\times 10^{-6},\mu =1\times 10^{-4}\right\}$, $\left\{\lambda =1.795,\gamma =1.5,\varphi =1\times 10^{-5},\mu =1\times 10^{-5}\right\}$, which are chosen to achieve optimal experimental results.

Moreover, we used four metrics: NMI, ACC, F-measure, and RI, to evaluate the proposed model.

Normalized Mutual Information (NMI):

NMI measures the similarity between the true labels and the predicted labels in the dataset. Based on our model, the NMI calculation method for this experiment is as follows:(26)$$MI=\sum_{i=1}^{C}\sum_{j=1}^{K}\frac{A_{ij}}{N}\log_{2}\left(\frac{{NA}_{ij}}{C_{i}K_{j}}\right).$$

In this function, $A_{ij}$ represents the intersection between the true labels and predicted labels in the dataset, $C_{i}$ is the number of experimental samples, $K_{j}$ is the number of samples in the *j*-th class, and *N* is the total number of experimental samples. Then, the entropy for each class is calculated:(27)$$H\left(C\right)=-\sum_{i=1}^{C}\frac{C_{i}}{N}\log_{2}\left(\frac{C_{i}}{N}\right),\;H\left(K\right)=-\sum_{j=1}^{C}\frac{K_{i}}{N}\log_{2}\left(\frac{K_{i}}{N}\right).$$

Finally, we obtain the calculation formula of NMI for this experiment:(28)$$NMI=\frac{2\cdot MI}{H(C)+H(K)}.$$

Accuracy:

ACC can be obtained by using the Hungarian [41] algorithm to solve the problem of label mismatch in the data set. The specific calculation is as follows:(29)$$ACC=\sum_{i=1}^{n}\frac{c\_matrix[r\_d[i],c\_d[i]]}{N}.$$

In this function, c_matrix is computed by calculating the confusion matrix between the true labels, while r_d and c_d are the best matching values obtained after using the Hungarian algorithm.

F-measure:(30)$$P=\frac{num_{i}}{num_{p}},\;R=\frac{num_{i}}{num_{t}}.$$

Here, we denote ${num}_{i}$ as the number of samples between the predicted label and the true label, ${num}_{p}$ as the number of samples matching it in the predicted label, and ${num}_{t}$ as the number of samples matching it in the true label. Finally, we know that the calculation method of F-measure is(31)$$F=\frac{2\cdot P\cdot R}{P+R}.$$

Rand Index (RI):

During this experiment, the calculation method of RI index is as follows:(32)$$RI=\frac{A}{M}.$$In this function, *A* is the number of matching pairs between cluster labels and *M* is the number of all possible cluster label pairs.

### 4.2. Datasets and Comparative Algorithms

In many applications, data such as text, images, and videos can often be represented through multi-view features. In this study, we conduct experiments on four datasets of different modalities—see Table 2. Experiments are implemented in Python 3.8 with 32 GB RAM and a 2.5 GHz Intel i5 processor. This code is compiled using Visual Studio Code (version: 1.99.2) released by Microsoft Corporation (Silicon Valley, CA, USA).

In this experiment, we use four different real-world datasets to quantitatively evaluate the effectiveness of our model. Specifically, the ORL [42] dataset is a widely used dataset for face recognition, containing images of 40 individuals, with a total of 400 images, consisting of 40 directories, each representing a different person, meaning that each directory contains 10 images. The Reuters [43] dataset mainly consists of documents written in five different languages and translations for six common categories. In this dataset, English documents are treated as one view, and translations are provided in four different languages. In our experiment, 210 documents are randomly selected, with 42 documents per class. The MSRCV1 [44] dataset mainly consists of 210 images from 7 different categories; it includes feature extraction for 5 different types. The BBCSports [45] dataset consists of sports news articles corresponding to five themes, with each document associated with two different types of features.

We apply multiple metrics, including Normalized Mutual Information (NMI) [46], Accuracy (ACC) [47], F-measure [48], and Rand Index (RI) [49], to evaluate the performance of the proposed algorithm. Higher values indicate better performance.

We have conducted experiments on multiple datasets by comparing with five state-of-the-art models:

MSSC [50]: A multi-modal extension of sparse subspace clustering and low-rank representation algorithms is proposed, which achieves the robust clustering of multi-modal data by leveraging the self-expression property of each modality and enforcing shared representations across modalities, while also handling nonlinear data through kernelization.

DiMSC [34]: This method explores the complementary information between multiple views by introducing the HSIC standard, reducing redundancy and improving clustering accuracy.

LT-MSC [38]: A Low-Rank Tensor Constrained Multi-view Subspace Clustering (LT-MSC) method is proposed, which views the subspace representation matrices of different views as a tensor and introduces a low-rank constraint to effectively reduce redundancy and improve clustering accuracy.

LMVSC [51]: A large-scale multi-view subspace clustering method is proposed to handle linearly complex multi-view data.

MCLES [52]: A unified optimization method is proposed to achieve multi-view clustering in the latent embedding space.

### 4.3. Comparative Studies

Our method first projects all samples to a consistent dimensionality using low-rank embedding, followed by a latent multi-view learning fusion mechanism that maps information from different views into a unified semantic space. In contrast, the other methods fix the dimensionality of the embeddings.

Based on the data in Table 3, we have conducted a comparative analysis based on the following four metrics.

The NMI metric: The proposed LRE-LAMVSC method demonstrates excellent results across all four datasets, particularly in the ORL dataset, where the NMI value of LRE-LAMVSC reaches 94.4%, significantly exceeding the second-best method, i.e., DiMSC (94.0%). In the Reuters dataset, the NMI value of LRE-LAMVSC is 42.5%, which, although relatively low, still outperforms LMVSC (37.28%) and MCLES (35.79%).

The ACC metric: LRE-LAMVSC performs exceptionally well in the ORL and MSRCV1 datasets, with values of 83.9% and 93.8%, significantly surpassing other comparison methods. Although the ACC for LRE-LAMVSC in the Reuters dataset is 45.7%, which is lower than LMVSC (48.17%) and MCLES (49.47%), it still outperforms MSSC (44.5%) and DiMSC (40.0%).

The F-measure metric: LRE-LAMVSC consistently maintains high performance, particularly in the ORL and MSRCV1 datasets, where it is clearly superior to other methods. In the Reuters dataset, LRE-LAMVSC achieves an F-measure of 51.3%, showing a certain improvement compared to LMVSC (48.89%) and MCLES (47.04%).

The RI metric: LRE-LAMVSC also performs outstandingly, particularly in the ORL and MSRCV1 datasets, where it significantly outperforms other methods. Although the RI value for LRE-LAMVSC in the Reuters dataset is relatively low, it still exceeds that of MSSC (67.09%) and DiMSC (67.49%).

Overall, the proposed LRE-LAMVSC method demonstrates excellent performance across the four metrics. The advantage of the proposed method lies in its ability to efficiently integrate information from different perspectives while demonstrating high robustness in terms of clustering accuracy, quality, and consistency.

In comparison, the proposed method combining low-rank embedding with latent space outperforms the state-of-the-art methods on the robustness in handling outliers and noisy environments. In particular, the proposed method shows better consistency among different views with improved ability of generalization and robustness.

Figure 2 shows the comparisons between the LRE-LAMVSC method and the other state-of-the-art methods across four different evaluation metrics (NMI, ACC, F-measure, and RI) and four different datasets (ORL, Reuters, MSRCV1, and BBCSport). The results show that the proposed LRE-LAMVSC method demonstrates better performance in all datasets, particularly in terms of NMI and RI, and achieves the best performance on the ORL and MSRCV1 datasets, indicating the significant advantage in clustering consistency and information sharing. Therefore, the overall performance of the LRE-LAMVSC method surpasses that of the other comparison methods across multiple datasets, fully validating the effectiveness and potential for development in multi-view clustering tasks.

Figure 3 shows the visualization of the experimental results on the real dataset for both the proposed method and the other three methods. As can be seen, our method outperforms the other three methods. Especially when comparing to the MSSC method, the LRE-LAMVSC method shows a significant improvement in clustering accuracy, indicating that this method can effectively capture the underlying structure of the data, thus enhancing the overall performance.

Figure 4 shows a detailed analysis of the ACC metric of the LRE-LAMVSC method with varying λ and the comparisons with the DiMSC, MSSC, LT-MSC, LMVSC, and MCLES methods. For the ORL dataset, on the bottom left, the proposed method demonstrated a certain advantage. When the parameter λ changed to a moderate value, LRE-LAMVSC maintained a steady increase compared to the other three models, reaching a peak of 83.90%, indicating the stability and effectiveness of the method on the ORL dataset. In comparison, LMVSC, MCLES, DiMSC, MSSC, and LT-MSC outperform LRE-LAMVSC in the parameter range from 0.50 to 1.25, but do not surpass LRE-LAMVSC overall. This indicates that our method exhibits high generalization ability on the ORL dataset. For the results of the Reuters dataset, in the top right of Figure 4, the proposed method also maintains superior performance, especially when λ<1.00, where the ACC values of our method are higher than the other methods and reach a peak of 45.70%. The bottom left of Figure 4 shows the results of the MSRCV1 dataset. As can be seen, the proposed method shows excellent performance on the ORL and Reuters datasets, with an ACC value improvement of 13.38% by comparing to the second-best model. The bottom-right of Figure 4 shows the results on the BBCSport dataset. As can seen, the proposed method shows minimal difference when compared to the other methods. But, for the comparison with DiMSC, our approach shows better performance.

Overall, the LRE-LAMVSC method outperforms other comparison methods on multiple datasets, especially on the ORL and Reuters datasets, where the ACC metric is improved significantly over the other methods. To sum up, the proposed LRE-LAMVSC shows significant advantages on the ORL and Reuters datasets: the proposed LRE-LAMVSC method maintains a high accuracy for four different metrics on different experimental datasets, demonstrating competitive generalization ability and robustness. The method also performs stably across multiple datasets and different parameter settings, which show the potential to perform effectively in various application scenarios. In the proposed LRE-LAMVSC method, a and b are two key parameters that control the low-rank constraint and noise matrix regularization, respectively. These parameters play a crucial role in balancing the model’s data fitting ability and overfitting prevention.

### 4.4. Discussions on the Comparisons with the Existing Methods

The core of the proposed LRE-LAMVSC method is to unify the latent features of multiple views through low-rank embedding, which significantly improves the consistency and anti-interference of cross-view learning. As can be seen in Table 3, the comparisons between the LRE-LAMVSC method with MSSC, LT-MSC, DiMSC, LMVSC, and MCLES on the ORL dataset show that the NMI (94.40%) and ACC (83.90%) of the LRE-LAMVSC method were better than those of MSSC (92.63% and 82.00%), DiMSC (94.00% and 83.80%), and LMVSC (73.30% and 47.75%). Therefore, our method shows more stability and consistency in noisy environments.

Moreover, previous works, such as the MCLES method, suffer falling into a local optimal solution during the process of the alternating optimization of the objective function, while the proposed LRE-LAMVSC method avoids this issue by using the ALM-ADM optimization framework. By comparing to the existing LMVSC methods, our proposed method on the Reuters dataset is better than LMVSC (NMI: 37.28%, ACC: 48.17%, F-measure: 48.89%) and MCLES (NMI: 35.79%, ACC: 49.47%, F-measure: 47.04%). The RI (80.5%) by our method is dominant among the compared methods, which shows that our model has certain advantages in cross-view consistency.

Furthermore, the proposed LRE-LAMVSC method not only retains the ability of traditional latent multi-view representation learning methods for complex data structures, but also fuses them into a unified latent representation in the process of low-rank embedding. From the results on the MSRCV1 dataset shown in Table 3, the ACC (93.38%) and F-measure (85.29%) of LRE-LAMVSC are better than other comparison methods, indicating that our model is more stable. Therefore, our method shows higher accuracy and consistency among noise and heterogeneity views in the MSRCV1 dataset.

## 5. Conclusions

This paper proposes a latent multi-view representation learning algorithm based on low-rank embedding to address the challenges of high-dimensional heterogeneity and cross-view association modeling in multi-view data analysis. Unlike traditional methods that rely on explicit view partitions, the proposed approach integrates low-rank matrix factorization with latent space mapping, creating an association learning framework with noise robustness. The contributions include (1) the design of a latent space representation model that effectively captures cross-view structural consistency through low-rank constraints; (2) the introduction of a collaborative iterative optimization mechanism for the projection matrix and adaptive view weights, employing an alternating direction optimization algorithm to jointly solve the feature embedding and weight assignment problems. Experimental results on the ORL and Reuters datasets benchmark demonstrate that the proposed method significantly outperforms existing state-of-the-art algorithms in terms of clustering, with the RI improving by 13% and Normalized Mutual Information increasing by 0.4%, alongside notably better stability in noisy scenarios. Future work will focus on exploring nonlinear kernel space extensions and deep neural network fusion strategies to enhance the modeling capability for complex nonlinear view relationships.

## Figures and Tables

**Figure 1 sensors-25-02778-f001:**
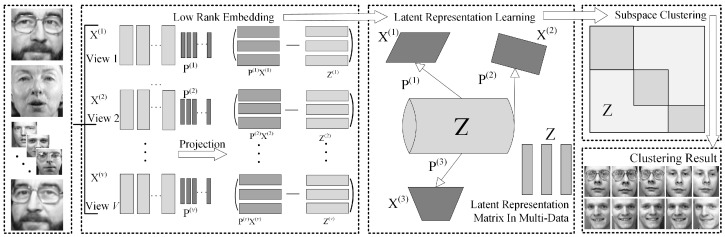
The framework of a novel Low-Rank Embedded Latent Multi-View Subspace Clustering (LRE-LAMVSC). We first pre-process the dataset by dividing into matrices of *v* different views. Then, we reconstruct the data using a low-rank embedding approach, which allows us to robustly reveal the underlying relationships between images. Next, we map the data from each view to a low-rank space, ensuring that the representations in this space have similar structures and shared latent information. Furthermore, these data representations are embedded into a shared latent space using the corresponding mapping matrix P. Finally, we employ an iterative strategy and the Augmented Lagrangian Multiplier method (ALM-ADM) to compute the optimal solution for LRE-LAMVSC.

**Figure 2 sensors-25-02778-f002:**
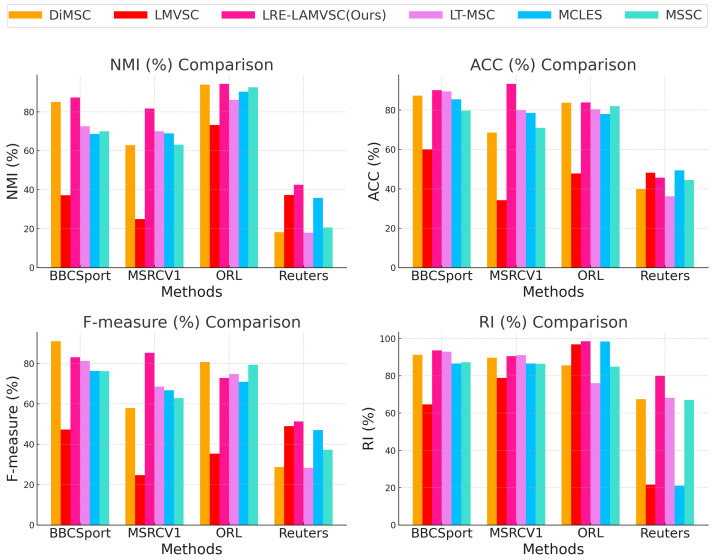
Comparison with the MSSC [50], DiMSC [34], LT-MSC [38], LMVSC [51], and MCLES [52] methods on four datasets, which are evaluated with four metrics: NMI, ACC, F-measure, and RI.

**Figure 3 sensors-25-02778-f003:**
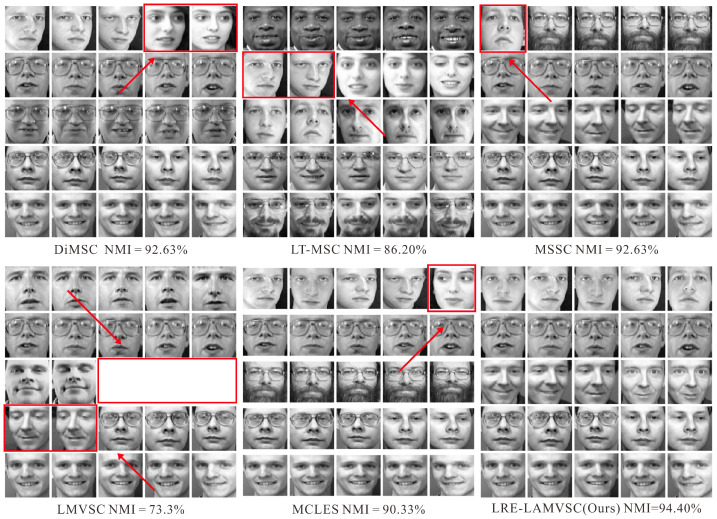
The output results of the proposed method and the state-of-the-art methods, including DiMSC [34], LT-MSC [38], MSSC [50], LMVSC [51], and MCLES [52]. The red arrows and boxes indicate the clustering errors.

**Figure 4 sensors-25-02778-f004:**
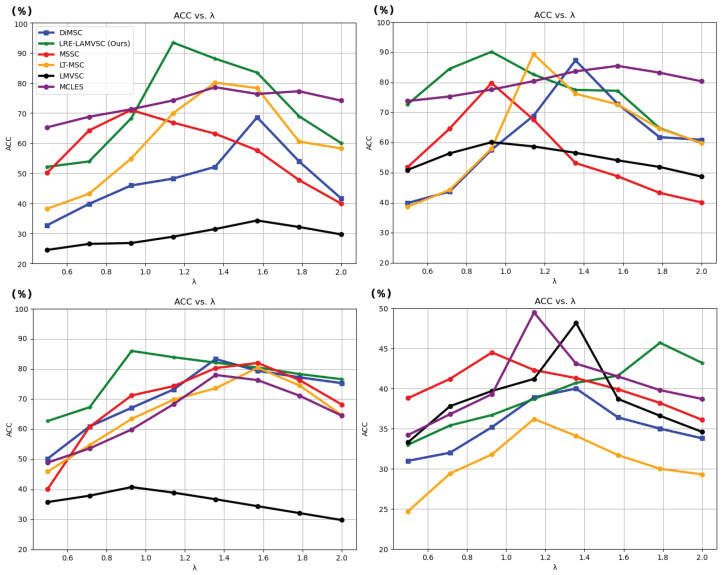
The comparisons among the proposed approach and the state-of-the-art methods, including LRE-LAMSC, DiMSC [34], LT-MSC [38], MSSC [50], LMVSC [51], and MCLES [52], with the ACC metric on the MSRCV1 dataset (**top left**), Reuters dataset (**top right**), ORL dataset (**bottom left**), and the BBCSport dataset (**bottom right**).

**Table 1 sensors-25-02778-t001:** Key notations in this work.

Notation	Explanation
$X^{\left(v\right)}$	the feature matrix of the *v*-th view
$P^{\left(v\right)}$	the projection matrix of the *v*-th view
$Z^{\left(v\right)}$	low-dimensional embedding across different views
$E^{\left(v\right)}$	error matrix
λ,γ,φ	regularization parameter
μ	penalty parameter
$Y_{1},Y_{2},Y_{3}$	lagrange multiplier

**Table 2 sensors-25-02778-t002:** Detailed information of the experimental datasets.

Datasets	Samples	Classes	Views
ORL [42]	400	40	3
Reuters [43]	2000	5	2
MSRCV1 [44]	210	7	5
BBCSport [45]	282	4	3

**Table 3 sensors-25-02778-t003:** Comparisons with the MSSC [50], DiMSC [34], LT-MSC [38], LMVSC [51], and MCLES [52] methods. Higher values indicate better performance.

Datasets	Methods	NMI (%)	ACC (%)	F-Measure (%)	RI (%)
ORL	MSSC	92.63	82.00	79.35	84.94
	DiMSC	94.00	83.80	**80.70**	85.60
	LT-MSC	86.20	80.40	74.80	76.10
	LMVSC	73.30	47.75	35.36	96.85
	MCLES	90.33	78.00	70.91	98.46
	LRE-LAMVSC (Ours)	**94.40**	**83.90**	72.80	**98.60**
Reuters	MSSC	20.56	44.50	37.23	67.09
	DiMSC	18.21	40.00	28.68	67.49
	LT-MSC	17.93	36.20	28.29	68.16
	LMVSC	37.28	48.17	48.89	21.73
	MCLES	35.79	**49.47**	47.04	21.07
	LRE-LAMVSC (Ours)	**42.50**	45.70	**51.30**	**80.05**
MSRCV1	MSSC	63.10	70.99	62.87	86.54
	DiMSC	62.87	68.57	57.92	89.72
	LT-MSC	70.04	80.00	68.48	**91.12**
	LMVSC	24.95	34.28	24.74	78.93
	MCLES	68.89	78.57	66.77	86.68
	LRE-LAMVSC (Ours)	**81.70**	**93.38**	**85.29**	90.60
BBCSport	MSSC	69.96	79.78	76.13	87.27
	DiMSC	85.11	87.32	**91.02**	91.32
	LT-MSC	72.56	89.43	81.19	92.91
	LMVSC	37.18	60.11	47.27	64.73
	MCLES	68.70	85.48	76.32	86.68
	LRE-LAMVSC (Ours)	**87.31**	**90.17**	83.09	**93.70**

## Data Availability

The data and experimental code of this article are available from the corresponding author upon request. For privacy reasons, the experimental code is not made public. During the writing process of this article, the author reviewed and edited the content as needed and takes full responsibility for the content of the publication.

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
