# Peer review of "A Novel Low-Rank Embedded Latent Multi-View Subspace Clustering Approach"

_sensors, 2025, doi:10.3390/s25092778_

Round 1
Reviewer 1 Report
Comments and Suggestions for Authors
The article is devoted to the analysis of sensor systems, but its relevance is not fully disclosed. The introduction does not contain a clear explanation of what scientific or practical gaps this study fills.
First of all, I recommend expanding the introduction section by adding relevant works on optimization:
Sakovich, N., Aksenov, D., Pleshakova, E., & Gataullin, S. (2024). MAMGD: Gradient-based optimization method using exponential decay. Technologies, 12(9), 154.
Comparative analysis with other methods, for example, with more traditional machine learning algorithms or modern deep learning models, is not carried out.
No justification is given for why the chosen method is preferable to alternative approaches.
Add comparative analysis with alternative methods to show the advantages of the proposed approach.
Describe the data characteristics in detail: measurement frequency, sensors used, features of their operation.
Provide numerical indicators of the method's efficiency, without limiting yourself to descriptive characteristics.
Author Response
Q 1: First of all, I recommend expanding the introduction section by adding relevant works on optimization:
Sakovich, N., Aksenov, D., Pleshakova, E., & Gataullin, S. (2024). MAMGD: Gradient-based optimization method using exponential decay. Technologies, 12(9), 154.
A: Thank you for the suggestion. While we focus on presenting the background, motivations and the novelty of this work in the Introduction section, we have explained the relation of the mentioned work in the Related Works section. See the red texts in the 1st paragraph of this section in Page 3.
Q 2: Comparative analysis with other methods, for example, with more traditional machine learning algorithms or modern deep learning models, is not carried out.
A: Thanks for the suggestion. We were aware of this and continued working on the comparisons after our initial submission. In this revised version, we have compared our method with another two methods including LMVSC and MCLES. The experimental results and corresponding discussions have been presented in Section 4, including Figure 3 and 4, and Table 3, and an extra discussion in Section 4.4. These newly added results have further shown that our method outperforms the existing baseline methods in multiple indicators, especially in data scenarios with high noise and strong inconsistency between views. See the red texts in Section 4.
Q 3: No justification is given for why the chosen method is preferable to alternative approaches.
A: Thanks for your comment. As being presented in the introduction, we have summarized three main insufficiency of the existing algorithms and proposed corresponding solutions by presenting the novelty of the proposed method. Specifically, the proposed method combines low-rank embedding representation with adaptive noise suppression mechanism, which has natural advantages in dealing with redundancy, noise interference and inconsistency between views in multi-view data. With additional comparative studies in the revised manuscript (see the red texts in Section 4), our method is more competitive for data modeling with complex structures. We believe these observations can better illustrate the uniqueness and applicability of our method.
Q 4: Add comparative analysis with alternative methods to show the advantages of the proposed approach.
A: Thank you for your comments. As being mentioned in the response to Q1.2, we have compared our method with another two methods including LMVSC and MCLES, which have further demonstrated the effectiveness and the efficiency of our method. See the red texts in Section 4.
Q 5: Describe the data characteristics in detail: measurement frequency, sensors used, features of their operation.
A: Thank you for your suggestion. We have added the detailed descriptions of the used dataset above Table 2. By following this suggestion, we have also added a specific paragraph below Table 2 to present each of the compared method.
Q 6: Provide numerical indicators of the method's efficiency, without limiting yourself to descriptive characteristics.
A: Thank you for the reviewer's valuable comments. We have added the computational complexity analysis and the descriptions in Section 3. See the red texts above Algorithm 1.
Reviewer 2 Report
Comments and Suggestions for Authors
The manuscript presents a latent multi-view representation learning model based on low-rank embedding that implicitly uncovers the latent consistency structure of data. It achieves robust and efficient multi-view feature fusion. Experimental results on multiple benchmark datasets demonstrate superior performance.
--Please discuss how this traditional multi-view learning relates to and differentiates from modern multi-modal learning, especially the multi-modal large language models.
--The problem definition needs to be described at the beginning of section 3 preferrably with a table of notations.
--The definition of "view" in multi-view learning can vary from study to study. It would be beneficial to summarize which data can be treated as a view in practical applications. E.g., in [R1], a data modality is defined as a view, whilst in [R2], a type of representation is defined as a view.
--The proposed approach is closely related to those proposed in [R2-R3]; please discuss what the differences are.
--Please add references to the comparison methods in Table 2.
--Figure 4 shows all methods are sensitive to the parameter lambda; how should the parameters lambda and others be chosen in practice?
--The datasets used in the experiments need to be described in more detail.
[R1] Cross-domain structure preserving projection for heterogeneous domain adaptation
Author Response
Q 1: Please discuss how this traditional multi-view learning relates to and differentiates from modern multi-modal learning, especially the multi-modal large language models.
A: Thank you for your valuable comments. In the Related Works section, we have added several key references to enhance the discussion on the relationship and difference between traditional multi-view learning and the modern multimodal learning. See the red texts in this Section. Moreover, we have also added a specific paragraph below Table 2 to present each of the compared method.
Q 2: The problem definition needs to be described at the beginning of section 3 preferrably with a table of notations.
A: Thank you for the helpful suggestion. We have added a new Table 1 to explain a list of the key notations. Inspired by this, we have added the detailed descriptions of the used dataset above Table 2, and added a specific paragraph below Table 2 to present each of the compared method.
Q 3: The definition of "view" in multi-view learning can vary from study to study. It would be beneficial to summarize which data can be treated as a view in practical applications. E.g., in [R1], a data modality is defined as a view, whilst in [R2], a type of representation is defined as a view.
A: Thank you for your comments. We have clarified the definition of "view" in Table 1, which is added to explain a list of key notations for presenting our method.
Q 4: The proposed approach is closely related to those proposed in [R2-R3]; please discuss what the differences are.
A: Thank you for the helpful suggestion. We have extended the descriptions of the mentioned references in the beginning of the 2nd paragraph (highlighted in red color) of the Related Works section.
Q 5: Please add references to the comparison methods in Table 2.
A: Thank you very much for your suggestions. We have added the corresponding references in the caption of Table 3, including the two newly added comparison methods.
Q 6: Figure 4 shows all methods are sensitive to the parameter lambda; how should the parameters lambda and others be chosen in practice?
A: Thank you for the suggestions. Based on the previous studies in [Auto-Weighted Multi-view Learning for Image Clustering and Semi-supervised Classification] by Nie et al., as also being shown in Figure 4, the parameter λ has a significant impact on the performance of different methods. The discussions and configurations of the other parameters are added in Section 4.1. See the red texts in the 1st paragraph in Section 4.1.
Q 7: The datasets used in the experiments need to be described in more detail.
A: Thank you for your suggestion. We have added detailed descriptions of the used dataset above Table 2. By following this suggestion, we have also added a specific paragraph below Table 2 to present each of the compared method.
Reviewer 3 Report
Comments and Suggestions for Authors
In this paper,the authors propose a latent multi-view representation learning model based on low-rank embedding by implicitly uncovering the latent consistency structure of data, which allows to achieve robust and efficient multi-view feature fusion. All in all, this paper is innovative and its presentation is clear. However, there are still some problems to be improved:
- Although the article mentions the importance of multi view learning, the motivation for combining low rank embedding with potential multi view learning is not clearly explained. For example, why choose low rank embedding as the core method? What are the unique advantages of low rank embedding in multi view learning? These questions have not been clearly answered.
- There are still some issues with the formatting of the formula, please carefully proofread the manuscript. For example, formula (1) and formula (2) do not have a unified layout for .
- In Results and discussion, only four benchmark datasets are used, mostly small-scale data (such as ORL, BBCSport), lacking validation for high-dimensional, large-scale, or more complex scenarios, and the universality of generalization ability is questionable. In addition, only the influence of parameter was analyzed, and the impact of key parameters such as (low rank constraint weight) and (noise matrix weight) in the objective function on the results was not mentioned.
- Some recent works should be discussed, including "Between/Within View Information Completing for Tensorial Incomplete Multi-view Clustering", "Focus More on What? Guiding Multi-Task Training for End-to-End Person Search" and "Manifold-based Incomplete Multi-view Clustering via Bi-Consistency Guidance".
- Although the article compares the performance with other methods, it lacks a detailed analysis of the advantages and disadvantages of different methods. For example, in which aspects do existing methods perform better? In what aspects does the LRE-LAMVSC method have significant advantages?
This paper looks promising but major revisions should be made before acceptance.
Author Response
Q 1: Although the article mentions the importance of multi view learning, the motivation for combining low rank embedding with potential multi view learning is not clearly explained. For example, why choose low rank embedding as the core method? What are the unique advantages of low rank embedding in multi view learning? These questions have not been clearly answered.
A: Thank you for your comments. In order to strengthen the proof of supporting the proposed model, we have conducted extra comparative studies by comparing to another two methods including MCLES and LMVSC (published in 2024). Furthermore, we have added a new sub-section, i.e., Section 4.4, to summarize the observations of all the comparative studies. It is also worthy to mention that additional comparative studies and analytical discussions have further demonstrated the effectiveness and the efficiency of our method. See the red texts in Section 4.
Q 2: There are still some issues with the formatting of the formula, please carefully proofread the manuscript. For example, formula (1) and formula (2) do not have a unified layout for .
A: Thank you for the comments. We have carefully proofread all the Equations, Tables and Figures throughout the manuscript.
Q 3: In Results and discussion, only four benchmark datasets are used, mostly small-scale data (such as ORL, BBCSport), lacking validation for high-dimensional, large-scale, or more complex scenarios, and the universality of generalization ability is questionable.
A: Thank you for the comments. Indeed the major contribution of this work is to study the potential of the proposed LRE-LAMVSC model. However, given the recent rapid developments on big models such as DeepSeek, we plan to extend the potential of this model on large-scale data by incorporating with big models.
Q 4: Some recent works should be discussed, including "Between/Within View Information Completing for Tensorial Incomplete Multi-view Clustering", "Focus More on What? Guiding Multi-Task Training for End-to-End Person Search" and "Manifold-based Incomplete Multi-view Clustering via Bi-Consistency Guidance".
A: Thank you for the suggestions. We have included the mentioned publications in our revised manuscript. See the red texts of the related works in Section 2.
Q 5: Although the article compares the performance with other methods, it lacks a detailed analysis of the advantages and disadvantages of different methods. For example, in which aspects do existing methods perform better? In what aspects does the LRE-LAMVSC method have significant advantages?
A: Thank you very much for your comments. We have added a specific part to explain all the 5 compared methods in the red texts below Table 2. Moreover, we have also added a new sub-section, i.e., Section 4.4, to summarize the observations of all the comparative studies.
Reviewer 4 Report
Comments and Suggestions for Authors
This paper proposes a novel multi-view subspace clustering method named LRE-LAMVSC, which integrates low-rank embedding representation and latent alignment mechanisms to project and fuse multi-view features into a unified semantic space. The approach enhances inter-view consistency and representation capability. Extensive experiments on four benchmark datasets (ORL, Reuters, MSRCV1, and BBCSport) demonstrate that the proposed method significantly outperforms several existing clustering algorithms. Furthermore, multiple experiments and robustness analyses confirm the effectiveness and stability of the model. However, the manuscript has several issues that need to be addressed before publication.
Suggestions for Improvement:
1.Page 6, from row 222: "In the 1, there exists a trivial solution for..." ---- "In the 1" is semantically unclear and lacks accompanying contextual information. A thorough proofread is recommended.
2.The caption of Figure 1 is simple. Each figure should be fully self-contained and informative.
3.Page 9, from row 291: "Section 4.1 provides a detailed description of the datasets and comparison methods." ---- There is no detailed description of the comparison methods in section 4.1. In addition, you should add more implementation details of the proposed method in the experimental part, such as parameter settings and evaluation metrics.
4.Add 1-2 recent methods from recent years in Table 2 for comparison.
5.The authors should follow up on the latest research progress. Some example peper's link are as follows:
https://doi.org/10.1016/j.patcog.2024.110709
6.A computational complexity analysis should be included to demonstrate the feasibility of the proposed method.
7.Ensure a uniform format for all references.
Author Response
Q 1: Page 6, from row 222: "In the 1, there exists a trivial solution for..." ---- "In the 1" is semantically unclear and lacks accompanying contextual information. A thorough proofread is recommended.
A: Thank you for the comment. We have carefully proofread all the labeling of Equations, Tables and Figures throughout the manuscript.
Q 2: The caption of Figure 1 is simple. Each figure should be fully self-contained and informative.
A: Thank you for the suggestions. We have added the detailed descriptions of Figure 1 in the caption.
Q 3: Page 9, from row 291: "Section 4.1 provides a detailed description of the datasets and comparison methods." ---- There is no detailed description of the comparison methods in section 4.1. In addition, you should add more implementation details of the proposed method in the experimental part, such as parameter settings and evaluation metrics.
A: Thank you very much for your suggestions. We have added the detailed descriptions of the compared methods, see the red texts below Tabel 2 in Section 4. The discussion and configuration of our parameters are supplemented in Section 4.1. See the red text in the first paragraph of Section 4.1.
Q 4: Add 1-2 recent methods from recent years in Table 2 for comparison.
A: Thank you for the suggestions. We have conducted extra comparative studies by comparing to two additional methods including MCLES and LMVSC, and the LMVSC method was published in 2024. It is worthy to mention that additional comparative studies have further demonstrated the effectiveness and the efficiency of our method. See the red texts in Section 4.
Q 5: The authors should follow up on the latest research progress. Some example peper's link are as follows:
https://doi.org/10.1016/j.patcog.2024.110709
A: Thank you for the suggestions. We have included the mentioned publications in our revised manuscript.
Q 6: A computational complexity analysis should be included to demonstrate the feasibility of the proposed method.
A: Thank you for the suggestions. The computational complexity analysis is conducted and presented in Section 3. See the red texts above Algorithm 1.
Q 7: Ensure a uniform format for all references.
A: Thank you for the suggestions. We have carefully proofread all the references, and also for all the labeling of Equations, Tables and Figures throughout the manuscript.
Round 2
Reviewer 1 Report
Comments and Suggestions for Authors
All comments have been taken into account